# Large-Scale Isolation of Milk Exosomes for Skincare

**DOI:** 10.3390/pharmaceutics16070930

**Published:** 2024-07-11

**Authors:** Xue Wu, Jiuheng Shen, Youxiu Zhong, Xian Zhao, Wantong Zhou, Peifen Gao, Xudong Wang, Wenlin An

**Affiliations:** China National Biotech Group (CNBG), Sinopharm Group, National Vaccine & Serum Institute (NVSI), No. 38 Jing Hai Second Road, Beijing 101111, China; wuxue2018a@163.com (X.W.); 18838002107@163.com (J.S.); zhongyouxiu@sinopharm.com (Y.Z.); zhaoxian11112021@126.com (X.Z.); tongwz0224@163.com (W.Z.); gaopeifen1996@163.com (P.G.)

**Keywords:** milk exosomes, skin penetration, barrier protection, skin whitening, anti-aging, soothing

## Abstract

Exosomes are small membrane vesicles in a cell culture. They are secreted by most cells and originate from the endosomal pathway. A variety of proteins, lipids, and genetic materials have been shown to be carried by exosomes. Once taken up by neighboring or distant cells, the bioactive compounds in exosomes can regulate the condition of recipient cells. Typically, producing exosomes in large quantities requires cell culture, resulting in high production costs. However, exosomes are abundant in milk and can be isolated on a large scale at a low cost. In our study, we found that milk exosomes can promote the synthesis and reconstruction of stratum corneum lipids, enhance skin barrier function, and provide greater protection for the skin. Furthermore, milk exosomes have anti-inflammatory properties that can reduce skin irritation, redness, and other symptoms, giving immediate relief. They also exhibit antioxidant activity, which helps neutralize free radicals and slows down the skin aging process. Additionally, milk exosomes inhibit melanin production, aiding in skin whitening. Ongoing research has uncovered the benefits of milk exosomes for skin improvement and their application in cosmetics, skin healthcare, and other fields, and these applications are continuing to expand.

## 1. Introduction

Exosomes, a type of vesicle, are released by cells with sizes ranging from 30 to 150 nm and share a similar structure and have similar properties to their parent cells without having organelles, as they are secreted by cells [1]. They contain similar components, such as RNA, DNA, phospholipids, cytoplasmic proteins, and membrane proteins [2]. Exosomes have been considered novel mediators of cell–cell communication, influencing a variety of biological processes in different conditions [3]. Compared to synthetic materials, exosomes demonstrate high biocompatibility and lower toxicity [4,5]. They can cross a variety of biological barriers because of the various surface proteins of exosomes, which can interact with integrins [6]. In addition, exosomes, after undergoing engineered modification, exhibit enhanced targeting capabilities and an improved ability to penetrate biological barriers [7]. Exosomes are considered effective delivery carriers for encapsulating as well as delivering both RNA and DNA to cells [8]. Recent research has shown that stem cell-derived exosomes could be adsorbed by human skin, increasing the levels of collagen I and elastin in the skin, highlighting the potential for the integration of exosomes in cosmetics or therapeutics [9]. However, the high production cost of stem cell exosomes hinders further application in cosmetics [10].

Milk exosomes (mEVs), abundant in milk, have been identified as promising alternatives [11]. Additionally, the abundance of casein in milk renders it unsuitable for the large-scale preparation of highly pure milk exosomes using traditional methods such as tangential flow filtration, which are typically employed to separate exosomes from cellular sources. There is an urgent need to develop a method for the large-scale, high-purity preparation of milk exosomes. Low immunogenicity, excellent biocompatibility, and the ability to penetrate the gastrointestinal barrier, which is possessed by mEVs, make them suitable carriers for oral delivery [12]. The presence of CD47 on the exosome surface prolongs the circulation time of milk exosomes, preventing recognition by macrophages [13]. Furthermore, mEVs possess a multitude of biological functional properties, including antioxidant capacity, immunoregulatory capability, and antimicrobial activity [14]. Critical roles are played by mEVs because of these functions in regard to human health, such as protection of the gut barrier via regulation of microbiota and bone/muscle metabolism [15]. However, there is extraordinarily little research on the regulation of function using mEVs.

In this study, large quantities of mEVs with high purity were prepared through acid pretreatment and could be taken up (Figure 1). mEVs show good biocompatibility, can be taken up by cells, and can cross the skin barrier. They showed improvement in the functioning of skin, including skin barrier protection, skin whitening, antioxidants, and skin-soothing in cells as well as animal experiments. These findings demonstrate that milk exosomes hold promise for their development and application in cosmetics.

## 2. Materials and Methods

### 2.1. Isolation, Purification, and Lyophilization of Exosomes

Exosomes from milk were isolated using a modified version of a previously reported method. Briefly, 10 mL of acetic acid was added to 10 L of nonfat fresh milk and incubated at room temperature for 10 min. Tangential flow filtration (TFF) with a 0.22 μm pore size was employed to eliminate large vesicles. The supernatant was then collected via centrifugation at 5000 rpm for 10 min. To achieve high purity milk exosomes, the supernatant was further filtered through 500 kDa TFF filters to remove particles smaller than the exosomes. Bacterial cells and other contaminants were removed from the filtered exosomes by filtration through a 0.22 μm filter membrane. The exosomes were then lyophilized overnight in a vacuum using a freeze-dryer (Epsilon 2-6D LSC, Osterode am Harz, Germany) and stored at 4 °C.

### 2.2. Transmission Electron Microscopy (TEM)

The lyophilized exosomes were resuspended in water and then placed on a carbon-coated copper grid for 10 min. Uranyl acetate was added to the samples, which were subsequently observed under a Thermo Talos 120C microscope (Waltham, MA, USA).

### 2.3. Zeta Potential of Milk Exosomes

The zeta potential of diluted milk exosomes (1:100) was measured using a Zetasizer Nano ZS90 dynamic light scattering instrument (Malvern, UK).

### 2.4. Protein Quantification

The protein concentration was determined by a Bicinchoninic acid (BCA) protein assay kit (Sigma-Aldrich, Castle Hill, New South Wales, Australia). BSA standard or milk exosome samples were transferred to a 96-well plate, and a working reagent was added. Detection was performed using a spectrophotometer (SpectraMax M5, Sunnyvale, CA, USA) at a wavelength of 562 nm.

### 2.5. Western Blot Analysis

Milk exosomes were electrophoresed using sodium dodecyl sulfate-polyacrylamide gel electrophoresis (SDS-PAGE gels, Bio-Rad, Hercules, CA, USA). The proteins were then transferred onto the nitrocellulose (NC) membranes. After blocking for 1 h at room temperature with phosphate-buffered saline containing Tween 20 (PBST) buffer with 5% (*m*/*v*) skim milk, the blot was incubated with a primary antibody. Following this, secondary antibodies were applied before detection with an electrochemiluminescent (ECL) reagent (Beyotime Biotechnology, Shanghai, China).

### 2.6. Nanoflow Analysis

The concentration and size distribution profiles of the milk exosomes were measured using a nanoflow system (NanoFCM Inc., Xiamen, China) according to the manufacturer’s protocol. Samples were diluted 1:100 in water and were to be measured by nanoflow. A quantity of exosomes of 2 × 10^8^ particles/ml was resuspended in 100 μL of water and then combined with 1 μL of SYTO-9. After incubation at 37 °C for 10 min, the samples were centrifuged to remove unincorporated dye from the labeled exosomes. The concentration and size distribution were determined using the nanoflow system.

### 2.7. Size-Exclusion HPLC Analysis

The exosome purity was measured by high-performance liquid chromatography (HPLC) using a TSKgel G6000PWXL (7.8 mm ID × 30 cm L, Griesheim, Germany) column connected to an HPLC system (LC-20A SHIMADZU, Tokyo, Japan). PBS buffer was used as the mobile phase, and the ultraviolet (UV) absorbance was detected at 280 nm.

### 2.8. Residual Water Content Measurement

The residual water content was measured using an 831 KF Coulometer (Herisau. Switzerland). After powering on the instrument, we selected the KFC measurement mode. We then used an analytical balance to weigh the total mass of the sample and container, denoted as M1. Then, we poured the sample into the titration cell of the instrument through the sample port. We placed the container back on the analytical balance and weighed it again to obtain the mass M2. We calculated the actual mass of the sample introduced into the reaction as M = M1 − M2. Then, we entered the measured mass of the sample, M (g), into the instrument. Once the instrument automatically balances, the value displayed on the screen will be the moisture content, X (ppm).

### 2.9. Bioinformatics Analysis

#### 2.9.1. RNA Isolation and High-Throughput Sequencing of Transcriptome

RNA extracted from milk exosomes was treated with the Invitrogen TRIzol LS reagent (Carlsbad, CA, USA) and then enriched for mRNA using oligo (dT) magnetic beads. The extracted RNA was then fragmented, and reverse-transcription was carried out using random primers to generate cDNA. Second-strand cDNA synthesis was performed by adding dUTP. Subsequently, polymerase chain reaction (PCR) amplification was carried out after the cDNA repair to create the final library. The library was evaluated for quality using an Agilent 2100 before being sequenced on an Illumina Hiseq4000 sequencer (San Diego, CA, USA).

#### 2.9.2. Protein Extraction and Digestion

Exosomes were lysed using SDT buffer, and the protein concentration was determined with a BCA assay kit (Bio-Rad, Hercules, CA, USA). The BCA protein assay kit (Sigma-Aldrich, Castle Hill, New South Wales, Australia) was used to measure the protein concentration. Bovine serum albumin (BSA) standard or milk exosome samples were transferred to a 96-well plate, and a working reagent was added. LC-MS/MS and data analysis were carried out by Wayen Biotechnologies (Shanghai, China).

### 2.10. Cell Culture

Hacat, NIH3T3, and B16 cells were cultured in Dulbecco’s Modified Eagle’s Medium (DMEM) (Invitrogen) supplemented with 10% Fetal Bovine Serum (FBS) (Invitrogen) and 1% penicillin and streptomycin at 37 °C in 5% CO_2_. RAW264.7 cells were grown in DMEM with 10% FBS and 1× Antibiotic-Antimycotic at 37 °C with 5% CO_2_.

### 2.11. Cellular Uptake of Exosomes

Hacat cells were seeded into a 35 mm confocal dish at an initial density of 2 × 10^4^ cells/dish and cultivated for 24 h prior to the experiment. pKH67 was chosen as the fluorescent dye to prepare 20 μL of pKH67-labeled milk exosomes, which were then added to the dishes. After incubation at 37 °C for 6 h, PBS was used to wash the cells 3 times. The cells were fixed in 4% formaldehyde for 10 min, followed by washing them 3 times with a washing buffer; 1 mL of PBS was added before the image was taken. The cellular uptake by Hacat was observed using a Zeiss LSM980 confocal laser scanning microscope (Oberkochen, Germany).

### 2.12. Cytotoxicity Assay

The cytotoxicity of the milk exosomes on RAW264.7 and NIH3T3 cells was assessed using standard CCK-8 assays. A total of 8000 cells were seeded onto 96-well plates and grown for 24 h. The cells were then treated with various concentrations of milk exosomes for 48 h. After that, 10 μL of CCK-8 was added to each well, and the absorbance values of the wells at 450 nm were measured on a microplate reader.

### 2.13. Melanin Content Analysis

B16 cells were seeded onto 24-well plates and grown for 24 h. The cells were treated with various concentrations of milk exosomes for 48 h. Following this, the supernatants were aspirated, and the cells were washed with PBS, treated with 0.25% trypsin at 37 °C for 3 min, and resuspended in a fresh culture medium. Subsequently, 120 µL of 1 M NaOH containing 10% dimethyl sulfoxide (DMSO) was added to each well and heated at 90 °C for 30 min. The absorbance values of the wells at 405 nm were measured using a microplate reader.

### 2.14. Determination of the Maximum Tolerated Concentration

mEVs were dissolved in solutions with final doses of 0, 0.625, 1.25, 2.5, 5, and 10 g L^−1^. Twenty fish, exposed to different concentrations of mEVs were tested in a 96-well plate. The temperature, pH, and dissolved oxygen were checked daily. The number of dead and teratogenic fish was recorded at 48 h to measure the mortality and teratogenicity.

### 2.15. Melanin Synthesis Inhibition Assay in Zebrafish

Zebrafish embryos were collected and arranged into a 12-well plate, with each well containing 8 embryos and an embryo medium. The experiment included a blank control group, a positive drug group (arbutin) (3 g/L), and milk exosomes (1 g/L), with three biological replicates for each group. Each replicate contained 8 fish fry, and 5 mL of the working solution for each concentration group was added to each well. The test groups were incubated for 24 h in a constant temperature incubator at 28.5 ± 0.5 °C. The embryos were dechorionated using a 1 mg/mL solution of streptomycin protease. Following dechorionation, the embryos were fixed with 3% methylcellulose and observed and photographed under a stereomicroscope. While photographing, the zebrafish were positioned with their heads to the left, ventral side down, and their bodies were kept horizontal. All zebrafish photographs were taken under the same equipment and environmental conditions, ensuring consistent body positions. The melanin gray levels of each embryo were analyzed using Image J2 software, and the data from each group were then subjected to a test to determine the significant differences.

### 2.16. Tyrosinase Activity Analysis

After 48 h of treatment with mEVs, the B16 cells were washed twice with PBS, then lysed with PBS containing 1% of Triton X-100 and 0.1 mM phenylmethanesulfonyl fluoride. The lysed B16 cells underwent centrifugation to collect the supernatant. The supernatant was then incubated at 37 °C for 1 h after adding 0.1 M L-3,4-dihydroxyphenylalanine. Detection was performed using a spectrophotometer (SpectraMax M5, Sunnyvale, CA, USA) at a wavelength of 475 nm.

### 2.17. Measurement of Intracellular Reactive Oxidative Species

NIH3T3 cells were plated on 6-well plates at a density of 1.5 × 10^5^ cells per well and incubated overnight to allow for cell adhesion. Subsequently, the cells were treated with 5 μM CellROX for 30 min before being washed with PBS and resuspended in a fresh culture medium. The fluorescence intensity of the cells was then analyzed using flow cytometry equipment from BD (Franklin Lakes, NJ, USA).

### 2.18. Measurement of Reactive Oxidative Species in Zebrafish

Healthy 2-day-old zebrafish embryos were dechorionated with a 1 mg/mL solution of streptomycin protease. After dechorionation, the embryos were incubated for 30 min to ensure their normal survival. A blank control group, a positive group GSH at 0.1 g/L, and mEVs at 0.3 g/L were set up, each with three biological replicates. In each replicate, 8 fish fry were placed, and 5 mL of the working solution corresponding to each concentration group was added to each well. The test groups were placed in a constant temperature incubator at 28.5 ± 0.5 °C for 24 h. The ROS fluorescent probe 2-7-Dichlorodihydrofluorescein diacetate (DCFH-DA) (0.25 μM) was added, and staining was carried out under light protection conditions at 28.5 ± 0.5 °C for 50 min. After staining, the embryos were rinsed 2–3 times with culture medium and then anesthetized with 40 mg/mL of MS-222. Fluorescence from each embryo was collected using a Nikon inverted fluorescence microscope, and the fluorescence intensity of each embryo was analyzed using Image J software. A test for significant differences in average fluorescence intensity among the groups was performed.

### 2.19. RT-qPCR Quantitative Reverse-Transcription Polymerase Chain Reaction (qRT-PCR)

Quantitative real-time PCR analysis was obtained using the Power SYBR Green PCR Master kit (Applied Biosystem). The primer sequence used was as follows (5′ to 3′, Table 1):

The data were analyzed using the 2^−ΔΔCT^ method.

### 2.20. In Vivo Skin Delivery of Exosomes

The backs of ICR mice were shaved one day before applying pKH67-labeled milk exosomes topically. Care was taken to shave the mice without causing skin damage. The milk exosomes were applied to the skin surface of ICR mice for different time intervals (1, 2, 4, and 48 h) the following day. After application, any surface-adsorbed samples on the skin surface were removed using isopropyl alcohol. Skin biopsies were fixed by 4% formaldehyde and then prepared for cryosections. The skin sections were imaged using a Zeiss LSM980 confocal laser scanning microscope.

### 2.21. Dermatological Tests

#### 2.21.1. Multiple Times Skin Stimulation Test

One day before the experiment, the hair on both sides of the back of New Zealand rabbits was cut, and the hair removal area on both sides was about 3 cm × 3 cm, with the right side used as the application area and the left side as the control area. A total of 0.5 mL of samples was directly applied to the skin of the animal’s coating area, while the same amount of sterile water was applied to the control area once a day for 14 days. Starting from the second day on, the hair was cut before each application, and the residual was removed with pure water. After 1 h of observation, the average score of each animal per day was calculated according to the skin irritation/corrosiveness test outlined in the *Safety Technical Code for Cosmetics* (2015 edition). The skin irritation intensity was determined using the skin irritation/corrosiveness test in the *Safety Technical Code for Cosmetics* (2015 edition).

#### 2.21.2. Acute Skin Stimulation Test

The right eye of New Zealand rabbits was considered the administration eye, while the left eye was the control eye. A total of 0.1 mL of samples were dropped into the conjunctival sac of the right eye of the animal, the upper and lower eyelids were passively closed for 1 s, and the eyes were not rinsed within 24 h after the sample was dropped. The stimulation damage and recovery of the conjunctival, iris, and cornea of the animal were observed for 1 h, 24 h, 48 h, 72 h, 4 d, and 7 d after the drug administration. If there was no stimulus response after 72 h, the test could be terminated. If corneal or other eye irritation persisted beyond 7 days, additional observation for reversibility or irreversibility of the damage, t, generally not more than 21 days, was conducted. The control eye should be injected with the same amount of normal saline.

### 2.22. Statistical Analysis

The experiments were repeated a minimum of 3 times, and the values indicated the average standard deviation. A Student’s *t*-test was employed to assess the statistical significance. Statistical analysis was carried out using GraphPad Prism (GraphPad Software 10, La Jolla, CA, USA). 

## 3. Results

### 3.1. Large-Scale Isolation and Characteristics of Milk Exosomes

Bovine milk consists of a variety of extracellular vesicles as well as proteins. Caseins were highly abundant in milk and were similar in size to milk exosomes, making it difficult to isolate the exosomes with purity [16]. Phosphorylation is commonly found in most milk casein, endowing the surface with a negative charge state. It is critical to remove milk caseins for isolating milk exosomes. Isoelectric precipitation using acetic acid to aggregate is an essential strategy for this. Tangential flow filtration is then used to purify and concentrate the milk exosomes on a large scale. The resulting lyophilized milk exosomes are white, cake-like, with a high specific area that aids in solubilization (Figure 1A). The residual water content in lyophilized products is 1.7%, which is less than 3%. As can be seen in Figure 1B, the transparent liquid is observed when a lyophilized sample redissolved in aqueous media, displaying the typical Tyndall effect, demonstrating high dispersity, hydrophilicity, and solubility of the milk exosomes (Figure 1C). These results indicate that the lyophilization process is suitable for milk exosomes. The size and purity of prepared milk exosomes were evaluated by nanoflow. As evidenced in Figure 1D, the measured size of fresh milk exosomes is about 69.03 ± 13.66 nm. The average diameter of the freeze-dried samples was measured to be 68.07 ± 13.44 nm, which shows little change compared to the native milk exosomes (Figure 1E). Furthermore, the purity of milk exosomes was estimated by nanoflow through the detection of exosomes stained with fluorescence dye SYTO-9, which has a high affinity for DNA and exhibits enhanced fluorescence upon binding with the capacity to penetrate the cell walls [17]. The purity of the exosomes isolated from milk in aqueous media was 96.0%, demonstrating that the isolation method is suitable for the large-scale production of milk exosomes with good purity (Figure 1F). As presented in Figure 1G, the purity of the lyophilized milk exosomes was similar to that of the native exosomes. In addition, Appendix A shows that the appearance time of the native and lyophilized milk exosomes was 8.91 min and 8.71 min, with only a single peak, respectively. As can be seen in Appendix A, exosome markers were enriched in isolated mEVs by acid pretreatment, while the reticulum marker was not detectable in the mEVs. The number of exosome particles was 1.22 × 10^11^ particles/mL, and the ratio of the particle number to protein concentration was 6.49 × 10^11^ particles/mg. These results further indicate the isolated milk exosomes with high purity through pretreatment as well as TFF. Transmission electron microscopy was employed to assess the morphology of the native and lyophilized milk exosomes. As illustrated in Figure 1H(a,b), the transmission electron microscopy (TEM) image of milk exosomes shows highly dispersed typical vesicles, which is similar to the previous report. No obvious changes could be observed in the freeze-dried milk exosomes, further demonstrating that lyophilization did not affect the characteristics of the milk exosomes (Figure 1H(c,d)). Taken together, the abovementioned results indicate the successful scale-up isolation and lyophilization of milk exosomes.

### 3.2. Protection of Milk Exosomes on the Skin Barrier

The skin protects our body from external assaults like pathogens, xenobiotics, or UV irradiation, as well as prevents the loss of water and solutes. The skin is composed of two types of epithelia: simple and stratified epithelia. Simple epithelium can be found in sweat glands or vascular endothelia, where cells are arranged single-layered and tightly adhere to each other at tight junctions (TJs) to prevent the diffusion of solutes through the intercellular spaces. Tight junctions act as an effective barrier in simple epithelia and play a significant role in drug absorption through the skin. The skin is structured into layers from the inside out, including stratum basale (SB), stratum spinosum (SS), stratum granulosum (SG), and stratum corneum (SC). Tight junctions contain multiple components, including transmembrane proteins (claudin (CLDN), occludin (Ocln)), junctional adhesion molecules, and TJ plaque proteins (zonula occludens [ZO] proteins, multi-PDZ domain protein-1, and cingulin). The impact of milk exosomes on TJ function was evaluated in Hacat cells. Firstly, the cytotoxicity of the mEVs was measured by the CCK-8 assay. As can be seen in Figure 2A, there was no obvious toxicity when the concentration of mEVs reached 5 × 10^11^ particles/mL, proving the low cytotoxicity of mEVs. Then, the intracellular delivery of exosomes into Hacat cells is confirmed by a confocal laser scanning microscope (CLSM). As can be seen in Figure 2B, negligible fluorescence is observed in Hacat cells without treatment. In contrast, an obvious fluorescence is observed upon treatment with pKH67-labeled mEVs for 6 h, demonstrating successful uptake by Hacat cells. As shown in Figure 2C,D, the levels of Claudin-1 and Claudin-6 in Hacat cells which measured by qPCR (Table 1) decrease upon treatment with mEV for 48 h. Claudin-1 content treated with 1.25 × 10^9^ particles/mL increased to 2.3-fold compared to the control group. As can be seen in Figure 2D, the level of Claudin-6 increased with the concentration of mEV, with a 2.93-fold upregulation observed in the Hacat cells after the concentration rose to 1.05 × 10^10^ particles/mL. Additionally, the content of PPP2R2A (Figure 2E) in Hacat cells dramatically downregulates, suggesting that mEVs can enhance skin barrier function via the upregulation of TJ protein.

### 3.3. Skin Lightening of Milk Exosomes

Melanin, the primary pigment in human skin, plays a crucial role in shielding the skin from UV rays. Changes in melanin production can lead to hyperpigmentation, affecting both appearance and health. Therefore, melanogenesis inhibitors are valuable for medical and cosmetic purposes. Tyrosinase, a key enzyme in melanin synthesis, has an extensively studied role in regulating melanogenesis. This copper-containing enzyme is found in various organisms and is responsible for melanin production. Tyrosinase catalyzes two reactions in melanin formation: tyrosine hydroxylation and L-DOPA. These reactions result in the formation of melanin through the polymerization of o-quinones. Tyrosinase is believed to be the main regulator of melanin production, making it a prime target for interventions. Milk exosomes are gaining attention as potential tyrosinase inhibitors. The B16F10 melanoma cell line is commonly used as a model for studying differentiation, including melanogenesis. Melanin production in B16F10 cells can be induced by UV and α-MSH, with cell proliferation being a critical factor. The cell viability of mEVs is assessed using the CCK-8 assay, alongside the evaluation of melanogenesis levels under low cytotoxicity concentrations after treatment with milk exosomes and after treatment with varying concentrations of mEVs for 48 h. The results showed that mEVs up to 5.0 × 10^10^ particles/mL did not have a cytotoxic effect on the B16F10 cells when compared to the control group. Further analysis involved examining the effect of milk exosomes on melanin production in B16F10 cells. Treatment with different concentrations of milk exosomes led to a dose-dependent reduction in the melanin levels, with efficacy plateauing at concentrations exceeding 5.0 × 10^9^ particles/mL. Vitamin C was used as a positive control in these experiments.

Furthermore, zebrafish embryos were employed to assess the content of melanin. Prior to the experiment, the maximum tolerated concentration (MTC) of milk exosomes was evaluated using zebrafish embryos. The results demonstrate no mortality or deformities at concentrations of up to 1.25 g/L (Appendix A). In addition, multiple skin irritation tests confirmed the safety of the mEVs (Appendix A). The level of melanin in the milk exosomes group was down to 39.9% lower than in the control group, revealing that milk exosomes are available for hyperpigmentation disorders treatment. Figure 3B demonstrates the impact of mEVs on tyrosinase activity in B16F10 cells. The tyrosinase activity in cells is dramatically decreased with a rise in the concentration of milk exosomes as compared to the untreated group. When the concentration reached 1.0×10^10^ particles/mL, the tyrosinase activity decreased to 42.3%. The results are comparable with that of milk exosomes with a concentration of 5.0 × 10^9^ particles/mL, which is on the same level as melanin in B16F10 cells. Overall, the results indicate that milk exosomes may have the ability to reduce melanin production, making them a potential ingredient for skin-lightening products for use in the cosmetic industry.

### 3.4. Anti-Aging Effect of Milk Exosomes

During skin aging, oxidative stress and inflammation are the main intrinsic causes because they damage proteins, lipids, and DNA. Reactive oxygen species, which are the predominant free radicals in living organisms, can cause oxidative stress as well as inflammation when produced excessively. The antioxidant effect of mEVs was evaluated by determining the ROS content in NIH3T3 cells’ pretreatment with H_2_O_2_. First, the cell viability effects of milk exosomes on H_2_O_2_-stimulated NIH3T3 cells were assessed. The NIH3T3 cells were treated with increasing concentrations of milk exosomes, from 6.4 × 10^5^ to 1.0 × 10^10^ particles/mL. The results, presented in Appendix A, show that the control group and mEVs group did not have appreciable differences from each other. To investigate the impact of milk exosomes on the viability of NIH3T3 cells exposed to cellular stress as triggered by H_2_O_2_, the cells were treated consecutively with milk exosomes and H_2_O_2_. The result is showcased in Appendix A; cell viability in the control group was found to be decreased to 33.6% in the presence of 100 μM H_2_O_2_ for 4 h. However, with the pretreatment with 6.4 × 10^5^ particles/mL milk exosomes for 24 h, cell viability was restored to 75.4% and significantly inhibited H_2_O_2_-induced cytotoxicity.

To evaluate the ROS scavenging activity of milk exosomes, the fluorescence intensity of CellROX was assessed. As indicated in Figure 4A, the intensity of CellROX in NIH3T3 cells after the treatment with H_2_O_2_ obviously increased compared to without the treatment. In the presence of milk exosomes, the fluorescence intensity in the NIH3T3 cells pretreated with milk exosomes was strongly reduced, suggesting that milk exosomes played a protective role against oxidative stress in the NIH3T3 cells caused by H_2_O_2_. The ROS scavenging activity of milk exosomes was further assessed in zebrafish. Three independent groups were established: the control, GSH as a positive control, and mEVs. Each group was incubated with respective samples for 24 h. The signal of 2′,7′-dichlorofluorescein (DCF) served as an indicator of the content of ROS in the cells. As can be seen in Figure 4B, the fluorescence of DCF was assessed in zebrafish by the pretreatment with milk exosomes. The fluorescence significantly decreased with an intensity down to 13.9% compared to without milk exosomes exposure, suggesting that milk exosomes could protect zebrafish from H_2_O_2_-induced oxidative stress (Figure 4C). In conclusion, these results indicate that mEVs can function in a protective role against oxidative stress.

### 3.5. Soothing Effect of Milk Exosomes in Cells

To assess the soothing effect of mEVs on LPS-stimulated macrophages (Raw264.7 cells), using an in vitro model of inflammation, the biocompatibility of mEVs was first assessed using CCK-8 assessment. As illustrated in Figure 5A, cell viability exceeded 100% for various concentrations of mEVs, indicating favorable biocompatibility of mEVs at concentrations that exhibited pharmacological activity. Following treatment with mEVs, the levels of inflammatory factors, such as COX-2, IL-6, and TNF-α, as well as anti-inflammatory factors, were assessed in RAW 264.7 cells. Figure 5B illustrates that exposure to lipopolysaccharide (LPS) resulted in an increase in inflammatory cytokines in RAW 264.7 cells, with the expression of COX-2, IL-6, and TNF-α increasing by 1.12-fold, 5.77-fold, and 9.09-fold, respectively. Notably, mEVs at a concentration of 1.0 × 10^9^ particles/mL dramatically inhibited COX-2, IL-6, and TNF-α production by 69.9%, 50.6%, and 76.5%, suggesting that mEVs could attenuate the secretion of pro-inflammatory cytokines. Meanwhile, upregulated levels of anti-inflammatory factors such as Arg-1, Stat-6, and IL-10 were observed after treatment with LPS. Additionally, these anti-inflammatory factors further increased upon treatment with mEVs with a concentration of 1.0 × 10^9^ particles/mL (Figure 5C). These findings demonstrate that mEVs possess anti-inflammatory properties, as evidenced by their ability to reduce pro-inflammatory factor levels and enhance the anti-inflammatory factor contents, which could be utilized as a soothing agent in skincare agents.

### 3.6. Penetration Effect in Animal Models

Given the excellent properties of mEVs in regulating skin functions, which could be harnessed in cosmetics, skin penetration studies were conducted prior to practical application; before the experiment, an acute percutaneous toxicity test was implemented to evaluate the safety of mEVs. As described in Appendix A, both female and male rats exhibited a gradual weight gain throughout the experimental period. To assess the permeation of milk exosomes on the skin, SPF-grade C57 mice were utilized in this study. After removing the hair from mouse skin surfaces with a scraper, an equal volume of either blank samples or milk exosome-positive samples (1 × 10^11^ particles/mL, 20 μL/mouse) were smeared on the exposed skin surface. The green fluorescence of the mouse skin profile was observed by a fluorescence confocal microscope to judge the transdermal condition of the milk exosomes. Compared to cells subjected to a green fluorescent protein (GFP) (Figure 6B,E), green fluorescence was observed in the dermis of the mouse skin sections after treatment with mEVs for 1 h (Figure 6C,F). These results indicate that the milk exosome traversed the epidermal layer of the skin and entered the dermis after 1 h.

### 3.7. Mechanism of mEVs to Regulate Skin Function

To explore the mechanism of mEVs to regulate skin function, proteomes and transcriptomes were performed. As displayed in Appendix A, LC-MS/MS identified 1899 proteins in the milk exosomes. The top 50 proteins with the highest abundance are seen in Figure 7A. Analysis of these proteins and their protein interaction network revealed that nine proteins, such as ACTB, ALB, HSP5, CDH1, CD44, RHOA, ENO1, CDC42, ITGB1, and APOE, are involved in skin function regulation (Figure 7C). ACTB, RHOA, and CDC42 are clustered into tight junctions and pathways [18,19]. RHOA and CDC42 are recognized for their role in the regulation of melanogenesis [20]. CDH1, ENO1, and APOE are noted for their negative regulation ROS generation [21,22,23]. APOE, ALB, and CD44 in exosomes have been shown to exert a soothing effect through anti-inflammatory action [24,25,26]. Collectively, these findings suggest that mEVs can significantly improve the skin status mainly via the abovementioned proteins. In addition to these proteins, miRNA in mEVs is also involved in the regulation of skin conditions. Exosomal miRNAs have been reported as key components involved in a number of physiological and pathological processes. Transcriptomic analysis of the mEVs was conducted to further explore the potential mechanisms for improving skin function. A total of 442 miRNAs, identified in liquid chromatography–tandem mass spectrometry (LC-MS/MS), are shown in Appendix A. As can be seen in Figure 7B, the abundance of the top 50 miRNAs in mEVs is shown. It has been reported that miRNAs in milk exosomes, including miR-141-3p, miR-122a, miR-30a-5p, miR-205, miR-196a2, miR-320a, miR-16, miR-21, miR-7, miR-218, miR-107, miR-150, and miR-126, contributed to skin barrier recovery via regulating tight junctions [27,28,29,30,31,32,33,34,35,36,37,38,39,40,41]. Among these miRNAs, miR-130a, miR-125b-5p, miR-133b, miR-144, miR-27b, and miR-23b are capable of inhibiting melanogenesis and tyrosinase activity to achieve skin lightening. miR-141 and miR-181a/b/c could alter the ROS signaling pathway to decrease ROS levels for improvement in aged skin. The inflammation of cells is reduced by employing mEVs due to intrinsic miRNAs such as miR-155, miR-let-7, miR-124a, miR-21, miR-146a, and miR-150. These results demonstrated that mEVs have the capability of direct delivery of miRNAs toward skin cells, thereby exerting their biological functions.

## 4. Conclusions

Given the natural origin of milk exosomes and their demonstrated biocompatibility, they could be engineered to carry personalized combinations of active ingredients, paving the way for tailored skincare solutions that address individual skin concerns more effectively. The utilization of milk exosomes as a foundation for cosmetic formulations is aligned with the escalating consumer demand for sustainable and ethical beauty products. Milk exosomes’ origin from a renewable resource, coupled with their multifunctional attributes, positions them as a sustainable solution that could significantly reduce the environmental impact of skincare products. Milk exosomes’ capacity to penetrate the skin and deliver bioactive compounds opens up a new frontier in transdermal drug delivery systems. This is particularly beneficial for ingredients that are traditionally difficult to formulate or that have poor skin permeability. The findings also hint at potential therapeutic uses for milk exosomes in treating various skin conditions, such as inflammatory dermatoses, by modulating the immune response and promoting skin barrier repair. While this study has shed light on the potential of milk exosomes in skincare, further research is required to understand the long-term effects, determine the optimal concentrations for various applications, and clarify the mechanisms by which exosomal cargo influences skin health at the molecular level. The scalable and cost-effective isolation of milk exosomes could have significant economic implications for the cosmetics and personal care industry, offering a high-value ingredient sourced from an abundant and renewable resource. By enhancing skin barrier function and reducing inflammation, milk exosomes could contribute to a reduction in skin-related health issues, potentially leading to an improved quality of life and reduced healthcare costs associated with skin conditions.

## Data Availability

Data are contained within the article and Appendix A.

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
