# Peer review of "Large-Scale Isolation of Milk Exosomes for Skincare"

_pharmaceutics, 2024, doi:10.3390/pharmaceutics16070930_

Round 1
Reviewer 1 Report
Comments and Suggestions for Authors
The manuscript entitled “Large-scale isolated milk exosomes for skincare” describes a low cost method to obtain exosomes from milk and a series of in vitro experiments to evaluate the potential applications of such exosomes in cosmetic products.
Although the topic is interesting, the manuscript is hard to read. The authors use many abbreviations without explaining their meaning the first time they cite the abbreviation. In addition, English is poor making some sentences hard to understand.
The experimental protocol is not well organized and several experimental procedures are missing. For instance, at line 216 the authors state, “The residual water content in lyophilized product is 1.7%” but they do not describe how they determined the water content in the lyophilized product. At line 307, the authors report the results of experiments performed on zebrafish embryos but they do not describe how they performed these experiments in the materials and methods section. At line 346, the authors say “Three independent groups are established: control, GSH as positive control and mEVs” but they do not report these groups in the experimental section. At line 196, the meaning of the sentences “Twenty fish exposed to different concentrations of mEVs are tested in 96-well plate. Temperature, pH, and dissolved oxygen are checked daily. Number of dead and teratogenic fish are recorded at 48h to measure the mortality and teratogenicity” is obscure. The authors do not report results about teratogenicity of mEVs.
The authors claim (line 380) that milk exosomes “could be employed as a soothing agent in skincare agent”. The authors draw this conclusion because they observed that mEVs exhibited anti-inflammatory properties. The authors should demonstrate that the observed anti-inflammatory activity of mEVs is able to provide a soothing effect after application on the skin.
This reviewer believes that milk exosomes could be useful to develop new cosmetic products but their usefulness should be supported by in vivo experiments on human volunteers.
Comments on the Quality of English Language
English is poor making some sentences hard to understand.

Reviewer 2 Report
Comments and Suggestions for Authors
The manuscript presents a very interesting use of the isolation of exosomes from milk and their potential use in cosmetic applications. The topic is a hot topic and the manuscript suitable for publication. Some aspects to consider:
-Introduction should be rewritten in a clearer way, there are many points that are difficult to follow,
-Scheme 1 does not provide any information and should be removed
-Materials and method section is extremely detailed. However, it is not reflected in the main text.
-Are the exosomes unilamellar or multilamellar?
-Conclusions should be extender
Comments on the Quality of English Language
Grammar and spelling only present the need of minor changes.

Round 2
Reviewer 1 Report
Comments and Suggestions for Authors
The authors revised the manuscript according to the reviewer's comment but English has still to be improved. For instance at line 71, the sentence " large quantities of mEVs with high purity was prepared" should read "large quantities of mEVs with high were prepared. Please, revise English language throughout the manuscript.
Comments on the Quality of English Language
English language needs an extensive editing.
Author Response
Thank you for your continued guidance and for pointing out the areas in our manuscript that still require improvement, particularly with regards to the English language. We have taken your feedback seriously and have made comprehensive revisions to enhance the clarity and accuracy of our writing.
To address the specific example you provided, we have corrected the sentence at line 71 from "large quantities of mEVs with high purity was prepared" to the more grammatically accurate "large quantities of mEVs with high purity were prepared."
In addition, we have shared the revised manuscript with a group of peers for an additional round of language and content review to ensure that all sections are clear and concise.
The main changes in the revised manuscript have been highlighted in yellow. We believe that our revision has improved the quality of manuscript, and we hope that the manuscript is now acceptable for publication in Pharmaceutics.

Reviewer 2 Report
Comments and Suggestions for Authors
In my opinion, authors have addressed all the points and now the manuscript is publishable
Comments on the Quality of English Language
The language is suitable for a research article.
Author Response
Thank you for your continued guidance and for pointing out the areas in our manuscript that still require improvement, particularly with regards to the English language. We have taken your feedback seriously and have made comprehensive revisions to enhance the clarity and accuracy of our writing.
We have shared the revised manuscript with a group of peers for an additional round of language and content review to ensure that all sections are clear and concise.
The main changes in the revised manuscript have been highlighted in yellow. We believe that our revision has improved the quality of manuscript, and we hope that the manuscript is now acceptable for publication in Pharmaceutics.
